# Design of a gold clustering site in an engineered apo-ferritin cage

Chenlin Lu[1], Basudev Maity[2], Xue Peng[1], Nozomi Ito[2], Satoshi Abe[2], Xiang Sheng[3], Takafumi Ueno [2✉] & Diannan Lu [1✉]

Water-soluble and biocompatible protein-protected gold nanoclusters (Au NCs) hold great promise for numerous applications. However, design and precise regulation of their structure at an atomic level remain challenging. Herein, we have engineered and constructed a gold clustering site at the 4-fold symmetric axis channel of the apo-ferritin cage. Using a series of X-ray crystal structures, we evaluated the stepwise accumulation process of Au ions into the cage and the formation of a multinuclear Au cluster in our designed cavity. We also disclosed the role of key residues in the metal accumulation process. X-ray crystal structures in combination with quantum chemical (QC) calculation revealed a unique Au clustering site with up to 12 Au atoms positions in the cavity. Moreover, the structure of the gold nanocluster was precisely tuned by the dosage of the Au precursor. As the gold concentration increases, the number of Au atoms position at the clustering site increases from 8 to 12, and a structural rearrangement was observed at a higher Au concentration. Furthermore, the binding affinity order of the four Au binding sites on apo-ferritin was unveiled with a stepwise increase of Au precursor concentration.

[1] Department of Chemical Engineering, Tsinghua University, Beijing 100-084, China. [2] School of Life Science and Technology, Tokyo Institute of Technology, Yokohama 226-8501, Japan. [3] Tianjin Institute of Industrial Biotechnology, Chinese Academy of Sciences, and, National Technology Innovation Center of Synthetic Biology, Tianjin 300308, China. ✉email: tueno@bio.titech.ac.jp; ludiannan@tsinghua.edu.cn

Metal nanoclusters (NCs) are ultra-small particles below 2 nm in size, consisting of several to a hundred atoms[1,2]. Nature adopt various metal NCs for diverse functions, for example, $Mn_4CaO_5$ clusters in photosystem II systems[3], $Fe_7MoS_9C$ cluster in nitrogenase[4], and [2Fe–2 S], [4Fe–3 S], [3Fe–4 S], [4Fe–4 S] clusters in reductase[5], dehydrogenase[6], and hydrogenase[7].

Inspired by natural NCs, many efforts have been made to create artificial metal clusters. Among the noble metal NCs reported so far[8], Au NCs are of particular interest due to their unique electronic/optical properties and hold vast applications in catalysis[9–11], biomedical imaging[12–14], medical therapy[15–17], etc. In the last few decades, a variety of Au NCs has been developed, such as $Au_{18}(SC_6H_{11})_{14}$[18], $Au_{25}(PET)_{18}$[19], $Au_{30}S(S-t-Bu)_{18}$[20], $Au_{92}(TBBT)_{44}$[21], etc. Apart from small organic molecules, numerous proteins were utilized as protecting ligands or scaffolds. Xie et al. aroused the interest in using BSA for facile synthesis Au NCs with red fluorescent emission.[22], which motivated the emergence of a surge of studies on BSA-Au NCs[23–25]. Y. Lu et al. synthesized the first lysozyme-stabilized Au NCs and used it for $Hg^{2+}$ sensing[26]. Further, the spectrum of the capping proteins was extended to trypsin[27], pepsin[28], papain[29], etc. Most studies in this field have mainly focused on the preparation and applications of protein-protected Au NCs. It has been demonstrated that the protecting ligand has profound effects on the properties of Au NCs [30–32].

However, it is still a huge challenge to design and precisely control the structure and the coordination of the protein-protected Au NCs at an atomic level, which play vital roles in their functions. To date, only a few studies have provided insights into the structure of protein-Au NCs and attempted to explain how they form under the influence of protein.

Ferritin is a self-assembly protein cage consisting of 24 subunits with an 8 nm interior cavity. Due to the unique structural features, robustness, uniform size, and also inspired by its physical function of iron-storage, the apo-ferritin have been used as a building block for nanotechnology by accumulating various non-natural metal ions, like Co, Ni, Pd, Au, etc[33–35]. A trinuclear Pd cluster was fabricated at the 3-fold axis channel of apo-ferritin by replacing His114 with a non-coordinating alanine to demonstrate structural evidence of metal coordination using apo-ferritin as a scaffold[36]. A sub-nano $Au_{10}$ cluster was reconstructed at the 3-fold axis channel through reduction by $NaBH_4$ with ferritin crystal[37].

Our previous studies have shown that the 4-fold axis channel of apo-ferritin with a four-helix bundle structure, in which a Cd binding site with various coordination structures was constructed, has the potential to become a metal coordination center[38]. Herein, we designed and constructed Au NCs applying the 4-fold axis channel of apo-ferritin as a scaffold. The atomic structures of the Fr-Au NCs were unveiled by the protein crystallography. The X-ray crystal structure analyses combined with quantum chemical (QC) calculations reveal a unique clustering site up to 12 Au atoms supported by the designed cystine and histidine. Moreover, the structure of the gold nanocluster was precisely tuned by the dosage of the Au precursor. As the gold precursor concentration increases, the Au ions at the clustering site changes from 8 to 12. Besides, a rearrangement of $Au_{12}$ was observed at a higher Au concentration. Furthermore, the binding affinity order of the four Au binding sites on apo-ferritin was unveiled with a stepwise increase of Au precursor concentration. This study deepens the understanding of the design of artificial metalloproteins with gold clusters and advances the study of gold coordination chemistry with protein.

## Results and discussion

**Molecular design strategy and preparation.** The cage of recombinant L-chain ferritin from horse liver (rHLFr) shows a highly symmetrical structure with symmetric interfaces of the 2-fold axis, 3-fold axis, and 4-fold axis. Among them, 3-fold axis and 4-fold axis interfaces form unique channel structures (Fig. 1A, B). The assembled cage is connected by these two different types of channels. The 3-fold channel is formed at the interface of the third and fourth helix from three different subunits where charged residues as Asp122, Asp127, and Glu130 play essential roles in metal ions transportation[39–41]. The 4-fold channel is composed of four E-helices forming a parallel four-helix bundle with C4 symmetry (Fig. 1B, D). Contrary to the 3-fold channel, the 4-fold channel is mainly composed of hydrophobic residues (Leu161, Tyr164, Leu165, Phe166, Leu169, Leu171), and its physical role was barely explored. Recent studies have witnessed the possibility of metal binding at the 4-fold channel of ferritin from different sources. Cu ion[42] and Co ion[43] could bind to the His169 of M frog ferritin. Ag ion bound to the Met153 of bacterium ferritin PfFt[44]. Our previous studies have shown Cd binding at the 4-fold axis channel by introducing cysteine residues per monomer (L161C/L165C and R168C/L169C)[38], which demonstrated its potential as a workplace to design a gold cluster.

Figure 1D shows the channel configuration of the 4-fold axis channel. From the exterior to the interior of the cage along the 4-fold axis, there are three bottlenecks in the channel at residue Leu161, Leu165, and Leu169. The pore radius where Gln158 and Leu161 are located is less than the Van der Waals (VDW) radius of a water molecule (1.7 A)[45], suggesting impermeability of water and metal ions and verified by our recent molecular dynamics simulation study[46]. As a result, it is difficult for ions to enter into the four-helix bundle of the 4-fold channel directly from the exterior of the cage, and the transportation of ions into the cage through the 3-fold channel is a prerequisite before binding to the sites in the pore of the 4-fold channel (Fig. 1C). Our preliminary experiments (See Supplementary Note 1 and Supplementary Fig. S1) with gold ions using Cys mutants (161 C/165 C and 168 C/169 C) demonstrate the potential of the 4-fold channel of apo-ferritin as a site for constructing Au clustering. Another important insight is that the side chain of cysteine is relatively short, making the interaction between Au atoms bound on different monomers rather difficult. In addition, the binding mode presents less diversity with a common pattern of one cysteine binding to two Au atoms acting as a bridge. Therefore, to create an Au clustering site at the 4-fold channel, coordination residues with longer side chain and larger rotamer space are needed to build the interaction between Au atoms from different monomers. Previous reports have shown that in addition to cysteine, histidine is often involved in the coordination sphere of gold-protein composites[47–50]. Compared with cysteine, histidine exhibits more extensive side-chain rotamer space with Chi1 and Chi2 due to the longer side chain. Thus, in our design, four L169Cs were expected as the main fixing residues and four R168Hs as auxiliary coordinating residues (Shown in Fig. 1E), which yielded the mutant design of apo-R168H/L169C-rHLFr.

**Gold ions binding into apo-R168H/L169C-rHLFr cage and evaluation of the accumulation process.** The apo-R168H/L169C-rHLFr mutant was expressed in *E.coli* and purified following the same procedure in our previous report[37,38]. For gold accumulation, we used chloro(dimethylsulfide) gold(I) as the Au precursor. The reactions were performed at 20 °C in 50 mM Tris/150 mM NaCl (pH8.0) using 0 - 400 equiv. of Au precursor, followed by purification through dialysis and size exclusion chromatography. The solution stability of the apo-R168H/L169C-rHLFr Au(I) composites was checked by the measurement of UV-vis spectra at various time intervals of 1 day, 3 days, and 5 days

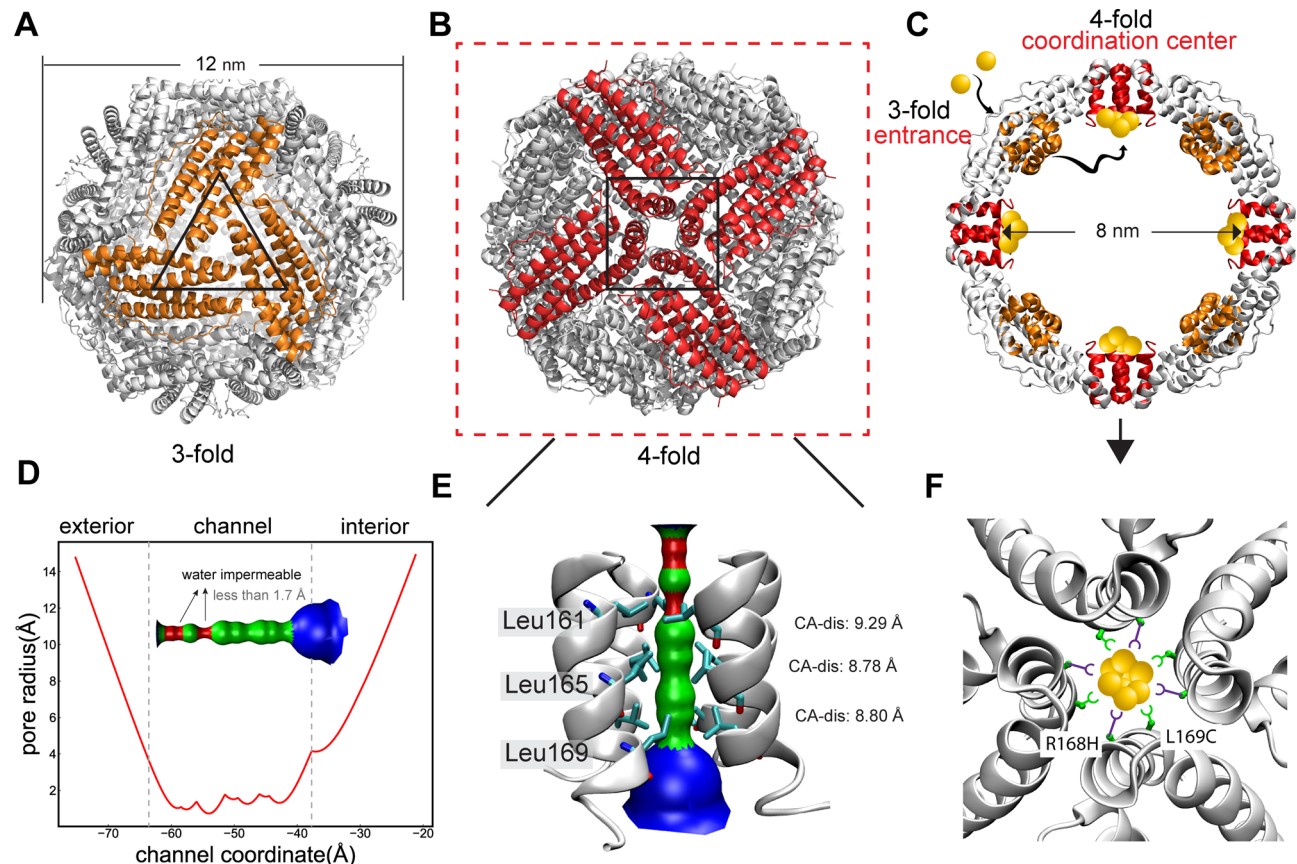

**Fig. 1 Structure of the ferritin and schematic presentation of the designed strategy.** Overall cage structure showing (**A**) 3-fold and (**B**) 4-fold symmetric facing front, respectively. **C** Interior cage structure of the ferritin cage showing the locations of the 4-fold channel (red helices) and schematic representation of the designed Au(I) accumulation pathways. **D** Pore radius along the 4-fold axis channel from the exterior to the interior of the rHLFr cage. **E** Hole configuration of the 4-fold axis channel with key residues contacting the channel interior. **F** Schematic presentation of the designed strategy for gold cluster formation at the 4-fold axis channel of the mutant apo-R168H/L169C-rHLFr. (Figures are prepared in PyMol and VMD).

after dialysis (Supplementary Fig. S8). No noticeable changes were observed for all the composites at different equivalents, which suggests that they are stable in solution at low concentrations (1–5 μM). If the samples undergo reduction, a remarkable adsorption peak at roughly 525 nm should be observed, as previously reported[51]. However, after a long time (over 2 weeks) the crystals and high concentration solution (~40 μM) utilized for crystallization gave nacked eye-visible color changes. The inductively coupled plasma mass spectrometry/ bicinchoninic acid assay (ICP-MS/BCA) quantitative analysis revealed that the number of the bound Au atoms are 4 ± 2, 46 ± 10, 87 ± 8, 158 ± 25, 203 ± 38 (Table S2) for Au(10 equiv.)-apo-R168H/L169C-rHLFr, Au(50 equiv.)-apo-R168H/L169C-rHLFr, Au(100 equiv.)-apo-R168H/L169C-rHLFr, Au(200 equiv.)-apo-R168H/L169C-rHLFr, and Au(400 equiv.)-apo-R168H/L169C-rHLFr, respectively.

X-ray photoelectron spectroscopy (XPS) analysis was further conducted to determine the oxidation state of Au ions in apo-R168H/L169C-rHLFr Au composites with different precursor concentrations (Supplementary Fig. S6). The high-resolution C1s spectrum can be decomposable into three carbons with different environments: the peak located at ~284.8 eV is assigned to the C–C, the peak at ~286.3 eV is ascribed to C–N/C–O, and the peak at 287.9 eV corresponds to C = O, which is consistent with the composition of amino acid residues[52,53]. The Au 4 f spectrum of apo-R168H/L169C-rHLFr Au composites (10, 50, 100, 200 and 400 equiv.) displays two peaks (Au 4f5/2, Au4f7/2) at binding energies of 88.24 eV, 84.30 eV; 88.15 eV, 84.50 eV; 88.12 eV,

84.49 eV; 88.39 eV, 84.72 eV, 88.28 eV, 84.62 eV; assigned to Au(I) with a slight right shift compared to the usual value of 88.50 eV for Au(I) compounds[54–56], which could be ascribed to Au(I)-His/Cys coordination[57,58]. The results suggested no oxidation or reduction of the Au(I) undergoes during the reaction between apo-R168H/L169C-rHLFr and gold source Me2SAuCl.

The Au accumulated ferritin cage was crystallized using the hanging-drop vapor diffusion method with (NH4)2SO4/CdSO4 as the precipitant. The X-ray diffraction data were collected, and the crystal structures were solved (See details in method section). Table S1 gives the crystal structure parameters of the apo-R168H/L169C-rHLFr. The resolution of apo-R168H/L169C-rHLFr is 1.5 Å. The crystal structures were determined using the structure of apo-rHLFr wild type (PDB code: 1DAT) as an initial model. The cage structures were rebuilt using the symmetry of the F432 space group. The mutation of R168H/L169C has a negligible effect on the conformation of the whole cage. The spherical structure of the mutant was found to be preserved as in apo-rHLFr wild type (PDB code: 1DAT). The crystal structure of all the Au composites were determined with the resolution of 1.9 Å following the same procedure as apo-R168H/L169C-rHLFr (Table S1). The spherical structures of the apo-ferritin cage were found to be conserved upon incorporating with Au(I) ions. The RMSD of C-alpha atoms of gold composites with reference to apo-R168H/L169C-rHLFr is within the range of 0.203-0.314 Å, which shows almost no difference upon gold binding.

The binding positions of Au ions were assigned by anomalous electron-density maps at 4σ and distinguished from Cd by

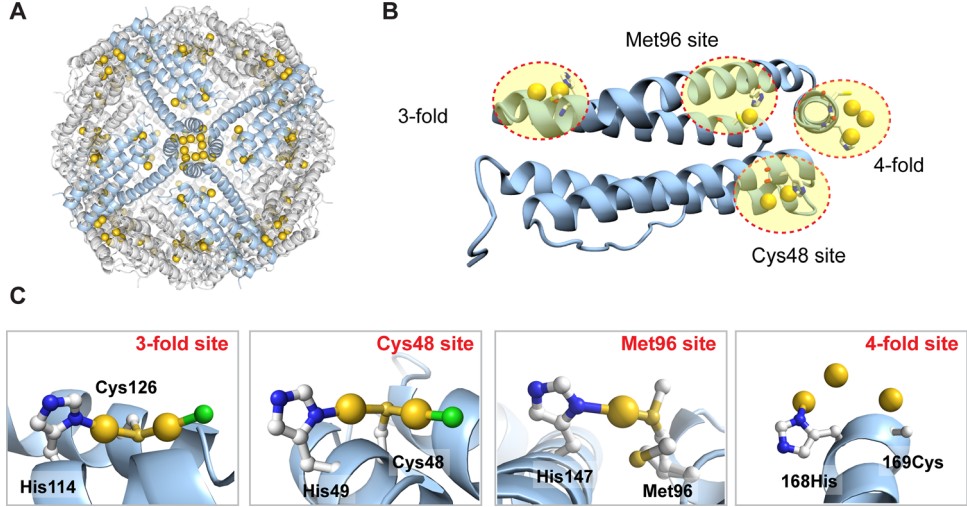

**Fig. 2 X-ray crystal structure of Au(400 equiv.)-apo-R168H/L169C-rHLFr containing the highest number of bounded Au. A** The whole spherical cage structure of Au(400 equiv.)-apo-R168H/L169C-rHLFr shows the immobilization of Au ions (yellow sphere) inside the cage. **B** Assigned Au binding sites on the monomer of Au(400 equiv.)-apo-R168H/L169C-rHLFr. **C** Coordination structures of the bounded Au in each site as shown in **B**. Figures were made in PyMol.

comparison with the control apo-R168H/L169C-rHLFr structure and anomalous scattering at multiple wavelengths of 1.15 Å and 1.00 Å (Supplementary Fig. S3). It was found that the Au ions were immobilized on the interior surface of the protein cage (Fig. 2A). The number of Au atoms estimated from all of the occupancies are 51.6, 98.4, 102.0, and 140.0, respectively. The results of 50 equiv./100 equiv. composites are close to those obtained from the ICP-MS/BCA quantitative analysis. In contrast, those of 200 equiv./400 equiv. composites are slightly smaller than ICP-MS/BCA results, which is attributed to non-specific Au binding on ferritin cage and could not be observed by crystallography due to the low occupancies. This might be a possible reason for observing an extended unassigned residual density in the 4-fold channel (Supplementary Fig. S4). The unassigned density might be additional Au positions with low occupancy which bind in a non-specific manner. The B-Factors/occupancies of Au atoms and bond distances between Au atoms and adjacent amino acids of Au composites at different equiv. at these four Au binding sites were listed in Table S3–S10. It is observed that there are totally four Au binding sites located on Au accumulated apo-R168H/L169C-rHLFr (Fig. 2B), including 3-fold site, Cys48 site, Met96, and designed 4-fold site. The close views of each Au binding site on Au composites with electron-density maps were shown in Supplementary Fig. S2. Among them, the Au binding sites at 3-fold, Cys48, and Met96 are consistent with the previous studies[37,59]. At these three sites, the Au ions coordinated to "S" of Cys or Met with His. A thiol (Cys) bridged dinuclear Au structures were observed at Cys48 and Cys126, which were supported by the surrounding His and a chloride ion (Fig. 2C). Interestingly, we observed that Au ions were bound at the 4-fold channel of our design. At the 4-fold site (L168H), 3 Au ions were assigned, one of which is coordinated to His168. The Cys169 was not assigned due to insufficient density. This suggests that the position of Cys169 might be dynamic during Au accumulation.

We studied the Au accumulation process along with conformational changes of coordinating residues at the 4 Au binding sites during the stepwise increase of the Au precursor concentration (Fig. 3). (1) For the Cys48 site: In the 100 equiv., 200 equiv. and 400 equiv. structures, two Au ions were bound, while only one Au ion binding was observed in 50 equiv. structure. First, one Au ion was captured by Cys48 in the 50

equiv. structure. Conformational changes were observed for Glu45 suggesting to be involved in the initial Au accumulation. This might be due to the electrostatic interaction with the positively charged Au ion with the Glu45 residue. Then, at 100 equiv. structure, an additional Au atom bound to Cys48. In the crystal structure, the density of Glu45 appeared weak, which suggests the possible dynamic nature of Glu45 during second Au ion accumulation. During this time, the His49 also changed its conformation and was involved in the second Au accumulation. The reaction of the ferritin cage with 200 equiv. gave a stable Cys48 bridged dinuclear Au coordination structure which did not change later. His49 coordinated to one of the Cys48 bounded Au ions. Such dinuclear structure remained the same as for 400 equiv. structure with increased occupancy. During such Au coordination, Arg52 changed its conformations along with Glu45 suggesting involvement in Au accumulation. (2) For 3-fold site: A Cys126 bridged dinuclear Au structure similar to the Cys48 site was observed for both 200 equiv./400 equiv. structures and the geometry is consistent with the reported results[37,60,61]. It is worth noting that in the 50 equiv. and 100 equiv. structures, we found that five/three Au ions were bound with relatively low occupancy at the 3-fold channel. First, at 10 equiv. precursor, two Au atoms were captured by Cys126 with occupancy 0.2 and 0.2 respectively. His114 did not directly coordinate with the Au at this stage. When the Au precursor is increased to 50 equiv., the presence of additional Au ions was observed. Cys126 bounded two Au ions had high occupancy whereas the other three Au ions has very low occupancy (~0.15). This indicates that these three Au ions are weakly bound and might be forming an intermediate clustering at the 3-fold channel due to aurophilic interaction during migration into the cage. Glu130 and His114 were found to support those Au ions as they are changing their conformations during the Au accumulation process (Supplementary Fig. S5). At 100 equiv. structure, we observed three Au positions. Cys126 bounded dinuclear Au structure was forming with support from His114. An additional Au ion was located with support from Glu130. At 200 equiv. structure, a stable Cys126 bridged dinuclear Au coordination structure was observed, His114 coordinate to one Au, and the other Au is coordinated by Cys126 and a Cl ion. The structure remained the same at 400 equiv. structure. This indicates the presence of additional Au ions at 50–100 equiv. structures are the

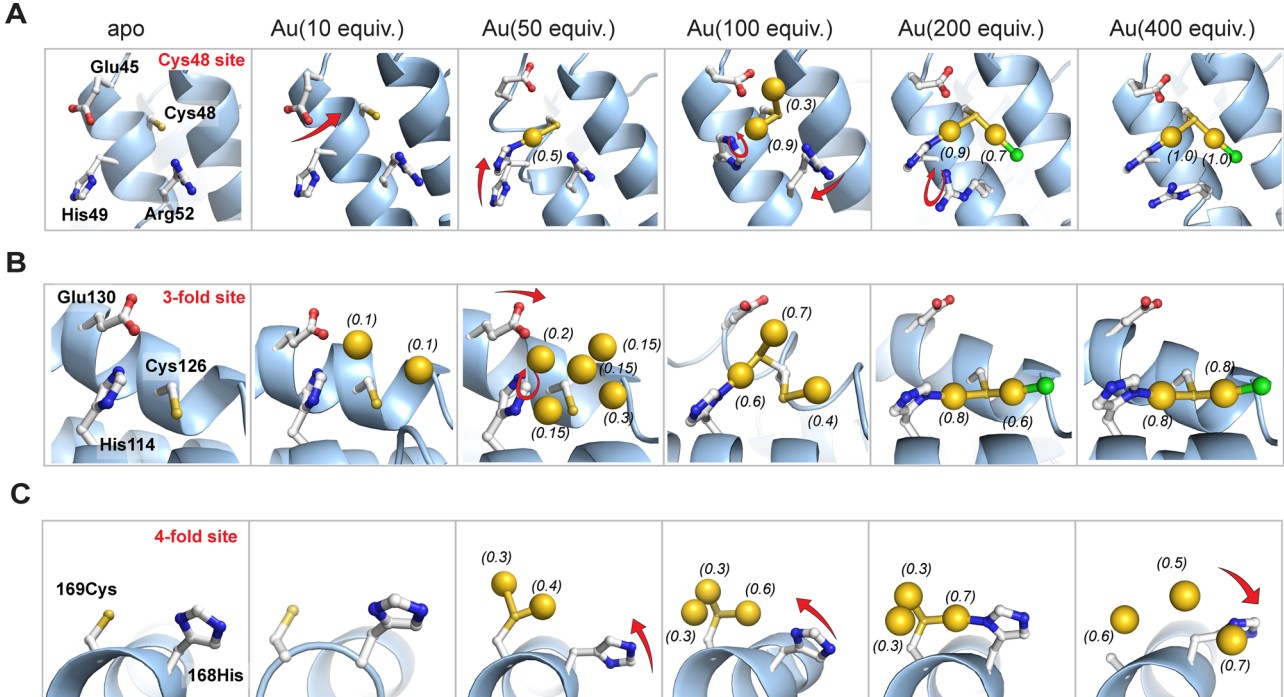

**Fig. 3 Accumulation process of Au ions into apo-R168H/L169C-rHLFr cage.** The conformational changes of key residues located at the Cys48 site (**A**), 3-fold site (**B**), and 4-fold site (**C**) with the stepwise increase of Au precursor concentration. The Au/Cl atoms are shown as yellow/green spheres. The occupancies of Au atoms were annotated next to the sphere models. Figures were made in PyMol.

intermediate structure of the Au accumulation process. It is also clear that Cys126, first captures the Au ions from the solution, the His114 and Glu130 helps to transport them into the cage. (3) For the 4-fold site: Two Au ions were bound in the 50 equiv. structure, and three Au ions were bound to the designed R168H/L169C per monomer in the 100 equiv., 200 equiv., and 400 equiv. structures. First, at 50 equiv. precursor, two Au atoms were captured by L169C with occupancy 0.4 and 0.3 respectively. Although the occupancies of these two Au ions are fractional, the Au–Au distance (2.72/3.13 Å) suggests possible aurophilic inter-action during initial Au accumulation at this site. R168H did not coordinate with the Au at this stage. When the Au precursor is increased to 100 equiv., R168H was observed to get close to Au atoms and the dinuclear Au positions remained the same as in 50 equiv. structure. At 200 equiv., an L169C bridged dinuclear Au coordination with coordination from R168H was observed. An additional Au ion was observed close to the dinuclear structure, however, the short Au–Au distance (1.8 Å) suggests that possibly they do not exist simultaneously. A further increase to 400 equiv. resulted in the rearrangement of 3 Au positions with His168 conformational changes. Higher Au concentration leads to the rearrangement. The side chain of L169C was not assigned due to insufficient electron density. (4) In the case of the Met96 site, the binding of Au by His147 and Met96 was only observed in the 400 equiv. structure and the geometry is similar to the previous reports[37,60].

The binding order of these four sites was further investigated by comparison of the structures along the axis with Au precursor increase (Supplementary Fig. S2). Au binding was first observed at the 3-fold site when the concentration increased to 10 equiv, which is followed by the 4-fold site and the Cys48 site when it reached 50 equiv. Additional density was obtained upon 200 equiv. with increasing occupancies (Table S3–S10). Only the Au binding at the Met96 site was witnessed at 400 equiv. It could be concluded that the binding order of these four sites is as follows: 3-fold site > 4-fold site > = Cys48 » Met96. It is noteworthy that a

similar binding pattern (Cys/Met)S-Au-N (His) maintains in all four sites, suggesting that in addition to cysteine, histidine also plays a crucial role in Au binding and can be exploited for the design of Au binding. This feature can be exploited for the further design of the Au binding and Au cluster structure. The 3-fold channel of the ferritin cage is commonly known for the entry of metal ions[62,63]. Supplementary Fig. S2 compares the Au binding of various composites starting from 10 equiv. to 400 equiv. First Au binding was observed at the 3-fold channel in 10 equiv. structure while other positions remained empty. In the 2nd stage, Au binding was observed at the 4-fold channel, and Cys48 which are located at the interior of the cage. This suggests that Au ions enter into the cage through the 3-fold channel and then, transfer to other binding sites. Recently, we also reported, how Au ions enter into the ferritin cage theoretically[46]. Therefore, our current studies prove the fact experimentally.

**Au clustering and precise regulation of the structure.** In the 4-fold site of apo-R168H/L169C, there are up to 3 Au ions positions per monomer as described in Fig. 3C. The reconstructed tetramer structure of the 4-fold channel based on the F432 symmetry reveals a total of 8–12 Au ions binding positions in the channel for 50-400 equiv. structures that were stabilized by the mutated residues R168H and L169C (Fig. 4A–D). The average Au–Au distances were found to be 2.6-3.4Å suggesting the presence of an Au cluster. The occupancies(B-Factors) of Au atoms and selected bond distances at the 4-fold site of the Au composites were summarized in Table 1. Since the occupancies of the Au ions in the 4-fold channel are fractional, it is hard to define the exact number of Au ions present at a time in the cluster. The structure of the multinuclear Au clustering site can be regulated by the dose of Au precursors. Fig. 4E–H shows the coordination structures in the 4-fold channel and the cluster configuration at different Au precursor concentrations. At low equiv. (50 equiv.), 8 Au ions positions were found where each R168C coordinates with

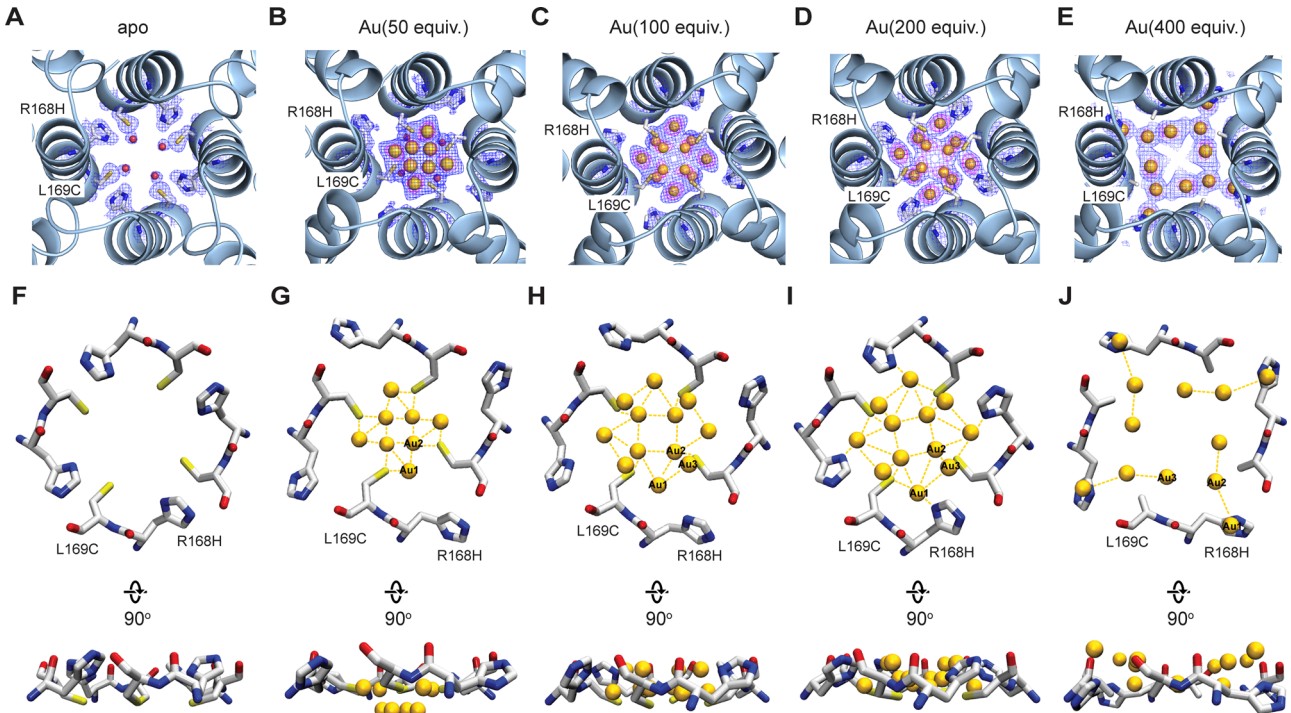

**Fig. 4 X-ray crystal structures of the assigned Au clustering site at the 4-fold symmetric channel in apo-R168H/L169C-rHLFr while using various Au precursors. A–E** Assigned positions of Au ions at the 4-fold axis channel of apo-R168H/L169C-rHLFr Au composites with different equiv. The selected 2Fo–Fc maps at 1σ and anomalous difference Fourier density maps at 4σ are shown in blue and magenta, respectively. Au(50 equiv.)-apo-R168H/L169C-rHLFr (**A**); Au(100 equiv.)-apo-R168H/L169C-rHLFr (**B**); Au(200 equiv.)-apo-R168H/L169C-rHLFr (**C**); Au(400 equiv.)-apo-R168H/L169C-rHLFr (**D**). **F–J** The coordination structure of each Au at the 4-fold axis channel of apo-R168H/L169C-rHLFr Au composites with different equivalents. apo-R168H/L169C-rHLFr (**F**); Au(50 equiv.)-apo-R168H/L169C-rHLFr (**G**); Au(100 equiv.)-apo-R168H/L169C-rHLFr (**H**); Au(200 equiv.)-apo-R168H/L169C-rHLFr (**I**); Au(400 equiv.)-apo-R168H/L169C-rHLFr (**J**). Figures were made in PyMol.

| | Occu./B.F.(Å²)/bond distance (Å) | | | |
|---|---|---|---|---|
| **Au atom/Bond** | **Au(50 equiv.)** | **Au(100 equiv.)** | **Au(200 equiv.)** | **Au(400 equiv.)** |
| Au1 | 0.4/40.86 | 0.6/46.27 | 0.65/31.10 | 0.7/51.83 |
| Au2 | 0.3/28.92 | 0.3/36.26 | 0.3/21.83 | 0.5/38.32 |
| Au3 | – | 0.3/37.99 | 0.3/23.76 | 0.6/42.03 |
| Au1-Sγ (L169C) | 1.84 | 2.59 | 2.08 | – |
| Au1- Nε(R168H) | >5 | 3.86 | 2.02 | 2.48 |
| Au2-Sγ (L169C) | 2.6 | 2.01 | 2.10 | – |
| Au3-Sγ (L169C) | – | 2.61 | 2.64 | – |
| Au2-Au2 | 2.16 | 2.71 | 2.73 | >5 |
| Au3-Au3 | – | 4.50 | 4.60 | 4.94 |
| Au1-Au2 | 3.13 | 2.96 | 3.15 | 3.43 |
| Au1-Au3 | – | 3.39 | 3.60 | >5 |
| Au2-Au3 | – | 1.84 | 1.81 | 4.12 |

**Table 1 B-Factors (B.F.)/occupancies (Occu.), and key bond distances at the 4-fold site of apo-R168H/L169C-rHLFr Au composites with different Au precursor concentrations.**

two Au ions to form a surface staple-like bonding structure as observed in Au₂₅(SR)₁₈[64] and Au₃₆(SR)₂₄[65] between thiolates ligand and surface gold atoms. The Au2-Au2 interaction, as well as the OW1, might play a crucial role in stabilizing the cluster structure. The central core of the 4-fold channel consists of 4 Au ions (Au2) from four different monomers with 30% occupancy (Fig. 4B, G). The short Au2-Au2 distance of 2.2 Å suggests that all the 4 Au2 might not exist simultaneously. In Au(100 equiv.)-apo-R168H/L169C-rHLFr, the 12 Au ions were found which did not

change in the 200 equiv. and 400 equiv. structures. The Au clustering structures for 100 equiv. and 200 equiv. are almost the same except for significant conformational changes of R168H in 200 equiv. The R168H is stabilizing the cluster through coordination to Au (Au1). In Au(400 equiv.)-apo-R168H/L169C-rHLFr, a significant change in the configuration of the Au cluster was observed with further conformational changes of R168H[37,60]. The results capitalized the significance of R168H in fine controllability of the unique Au clusters at the 4-fold channel by Au precursor concentration. Interestingly, no association of gold ions like 50-200 equiv. structure in the center of the pore channel was observed. At the same time, four additional gold ions were observed located in the slit of two monomers. Icosahedral or centered-icosahedral are common kernel structures observed in thiolate protected Au₁₂ and Au₁₃ nanoclusters[66,67], but in our case, the cavity size as well as coordinating residues are limited. So, the Au ions are forced to take the structure which become uncommon in literature. As demonstrated in Fig. 4, the Au12 cluster on the 4-fold site exhibits a two-layer layered morphology. It highlights the unique channel topologies that aided in the creation of this one-of-a-kind Au clustering site. Comparing all the four structures (50 equiv., 100 equiv., 200 equiv., and 400 equiv.) in Fig. 4(A–E), it was found that the clustering Au ions were rearranged with increasing Au precursor concentration which is significant for 400 equiv. structure. Cluster-to-cluster transformation in multinuclear Au(I) complex has been reported previously in which a 4 Au(I) cluster was transformed into 12 Au(I) cluster upon increasing concentration and vice versa[68]. Similarly, Au(I)-Au(I) interaction mediated solvent-induced Au₁₀ to Au₁₂ (cluster-to-cluster) transformation was observed[69]. Therefore, the Au structural transition in the 4-fold channel with

higher Au precursor concentration indicates the possible dynamic nature of the cluster similar to the previous reports. At higher gold precursor concentration, the equilibrium of metal-ligand(-protein) interaction is expected to be changed. Since amino acid side chains are flexible and can change their conformations according to the metal coordination structure, a more energetically favorable Au cluster structure was thus formed at 400 equiv. Figure 3c shows the conformational changes of L169H with increasing Au precursor concentrations which were reflected in the overall cluster structure in the 4-fold channel as well as structural transition.

We investigated the sequence of Au accumulation considering the coordination structure of Au (Figs. 3C, 4F–J and Supplementary Fig. S4) and the differences in electron-density map with the gradual increase of gold precursor concentration. It can be deduced that among Au1, Au2, and Au3 at the 4-fold sites, Au1 has the highest binding affinity, followed by Au2 and then Au3. As the concentration increases, the electron density of Au1 increases first, followed by Au2 and finally Au3 (Fig. 4, Supplementary Fig. S4). After reaching 200 equiv., further, an increase in concentration leads to a rearrangement of the gold cluster structure, where the electron density of Au1 and Au2 rearrange from the center to the edge. It is noteworthy that R168H shows a dynamic behavior in stabilizing gold clusters. At a low concentration of Au precursors, R168H is not involved in coordination. As the concentration increases, it gradually approaches the gold cluster and forms coordination bonds with Au2. The rearrangement of gold clusters caused by further concentration increment is also accompanied by conformational changes of R168H (side chain repack backward the channel center).

We further analyzed the Au clusters by detailed measurement of Au–Au bond distances. The 12 Au ions in the 100 equiv. and 200 equiv. structures could be classified into three types according to their different coordination spheres. The four Au1 were observed to coordinate to R168H and L169C with Au1-N and Au1-S distances of 2.02 Å and 2.08 Å, respectively. Similiar distances have been reported in $(CH3)_2Au(SR)$[70], $[Au(RSH)_2]$ $C1^+$[71], and $Na_3Au(S_2O_3)_2 \cdot 2H_2O$ complex[72]. Four Au2 located at the center of the unique cluster, each of which was stabilized by the L169C and two adjacent Au2, Au1, and thiolate groups of L169C. The Au2-S distance is 2.09 Å and is similar to Au1-S. The Au1-Au2 and Au1-Au3 distances are 3.14 Å and 3.59 Å, respectively, indicating the existence of the Au–Au aurophilic interactions[73–75] in this unique cluster as reported in (TPA)AuCl complex[76], $[Au_{10}(R\text{-}BINAP)_2(S\text{-}BINAP)_2(\mu3\text{-}S)_4]Cl_2$[77], Au10/ Au18 μ3-sulfido clusters[78], Octanuclear gold(I) alkynyl-diphosphine clusters[79], $[Au_2Ag_2(RI/\ RII)_4](RI = 4\text{-}C_6F_4I,\ RII = 2\text{-}C_6F_4I)$ bimetallic clusters[80]. The representative Au–Au, Au–S, Au–N bond distances observed in previous reports were summarized in Table S11–13. The Au3 was observed to bound Au1 and L169C. The Au3-L169C distance is 2.62 Å and is similar to Au1-S, Au2-S. However, the distance between Au3 and Au2 is surprisingly only 1.81 Å. Such a short Au–Au bond distance has not been reported before. It could be concluded that the coexistence of Au2 and Au3 on the same monomer is impossible. Further, QC calculation was applied to investigate the existence of Au–Au interactions in the cluster. Under the assumption of symmetric structure, when the cluster contains only four Au ions, there are three possible conformations, namely four Au1, Au2, or Au3 (Supplementary Fig. S7). All possible Au clustering configurations were extracted from the crystal structure as input geometries for QC calculations. The atomic coordinates of the optimized computational models were provided in Supplementary Data 1. The proportion of all possible conformations when there are four gold ions in the cluster were calculated based on their Gibbs free energy and Boltzmann distribution:

$$p_i = \frac{1}{Q} e^{-\Delta G_i/kT} = \frac{e^{-\Delta G_i/kT}}{\sum_{i=1}^{N} e^{-\Delta G_i/kT}}$$

Where $p_i$ is the probability of conformation i, $\Delta G_i$ is the Gibbs free energy of state i, $k$ is the Boltzmann constant, $T$ is the temperature of the system (equals to 298.15 K in our calculations), and $N$ is the number of all states accessible (equals to 3). As presented in Supplementary Fig. S7, when there are only four Au ions in the cluster, the conformation that contains four Au1 is predominant (100 %) among three accessible states due to the huge Gibbs free energy gap, which means that the electron-density map must not be an average of the three individual cases (4Au1, 4Au or 4Au3). The QC results combined with crystal structures might confirm the formation of the cluster rather than the average results of simple binding. A similar clustering was observed at the 3-fold site in 50 equiv./100 equiv. structures (Supplementary Fig. S5) with relatively low occupancy and attributed to initial transportation stage at low concentration, which also highlights the potential of constructing unique Au clustering site by integration of multiple simple binding sites into suitable channel structures.

We would like to note that although we assigned a total of 8–12 Au ions position in the 4-fold symmetric channel, there might be additional Au ions with very low occupancy and have non-specific binding which are difficult to identify by X-ray crystallography. This is also in agreement with the deviation of Au numbers determined from quantitative analysis (203 Au ions) to X-ray structure determination (140 Au ions). Even after assigning 8–12 Au ions based on the anomalous map at 4σ, there remained additional residual densities in Fo–Fc map at the 4-fold channel (Supplementary Fig. S4). We left unassigned those residual densities due to lack of any anomalous, they are very close to the Au ions and cover an extended area with an unusual shape. It is also possible that the residual density could be from water ligand or weakly bounded Au atoms or the coexistence of these two effects.

There are several reports on emissive protein stabilized gold nanoclusters[22,81]. To examine the photoluminescence property of our composites, we measured the UV-vis absorption spectroscopy (Supplementary Fig. S8) and Excitation Emission Matrix (EEM) spectroscopy (Supplementary Fig. S9) for all of the apo-R168H/ L169C-rHLFr Au(I) composites. No emission was observed for our ferritin-protected Au(I) clusters. The majority of the emissive gold clusters contained Au(0) oxidation state. Therefore, it is not unusual that our Au(I) clusters are non-emissive.

**Electronic structure unveiled by wavefunction analysis.** For more insight into the nature of the unique cluster formed at the 4-fold channel of our design apo-R168H/L169C-rHLFr, QC calculation and wavefunction analysis were conducted. The twelve Au ions and R168H/L169C were extracted from the crystal structure of Au(200 equiv.)-apo-R168H/L169C-rHLFr for QC calculation (see details in method section). During the geometry optimization, the positions of Au ions and the C-alpha atoms of R168H/L169C were fixed since their electron density was clearly observed. Firstly, the Atomic dipole moment corrected Hirshfeld (ADCH) charges[82] were calculated using the Multiwfn based on the wavefunction obtained from QC calculation. As shown in Fig. 5A–D, among three types of Au ions, Au1 and Au3 were observed to exhibit positive atomic charges (0.3538 e, 0.5715 e). In contrast, the four Au2 gold ions located at the center of the cluster yielded slight negative charges (–0.0005 e). A similar observation has been demonstrated by E. Kryachko et al. in the $Au_{20}$ cluster where the four apex Au atoms have negative

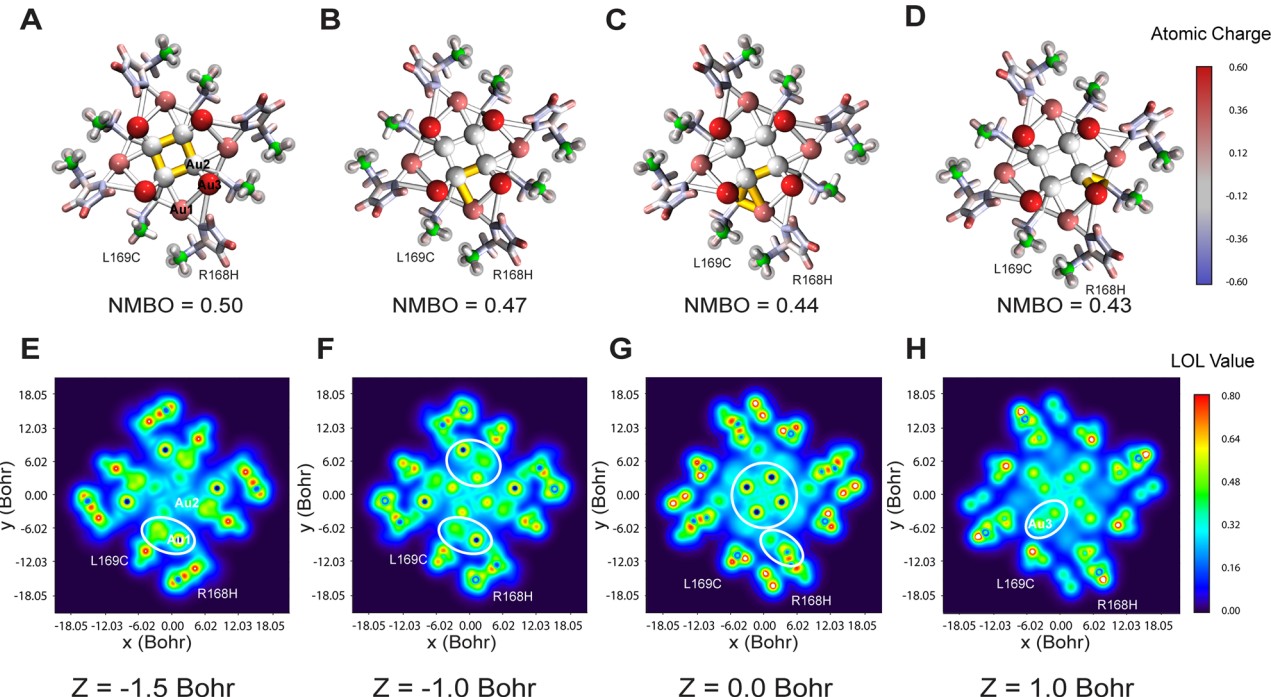

**Fig. 5 QC calculations and wavefunction analysis. A** The multi-center interaction between four Au2 atoms (highlighted in yellow). **B** The multi-center interaction between one Au1 atom, two Au2 atoms, and S atom of L169C (highlighted in yellow). **C** The multi-center interaction between one Au1 atom, two Au2 atoms (highlighted in yellow). **D** The multi-center interaction between one Au1 atom, one Au2 atom, and S of atom L169C (highlighted in yellow). The ADCH atomic charges are presented by the filled color on the licorice model in **A**–**D**, the C-alpha atoms of L169C/R168H were shown in green. **E**–**H** 2D localized orbital locator (LOL) color-filled map of Au12 at different Z-values. In the regions with large LOL values, electrons are more localized in them. Relatively, electrons can freely delocalize in such regions. Figures were made in VMD and Multiwfn.

Mulliken charges, and the other atoms have positive charges[83]. This particular atomic charge distribution attributes to the Au–Au and Au-residue interaction. Multi-center bond order (MCBO) analysis[84,85] was conducted to evaluate the multi-center interaction in the ferritin-protected $Au_{12}$ cluster quantitatively. In total, there are four pronounced multi-center interactions in this cluster (normalized MCBO > 0.4). The first one involves the four centrally located Au2 atoms (Fig. 5A, highlighted in yellow). The second one involves the S atom of L169C, one Au1, and two neighboring Au2 atoms (Fig. 5B, highlighted in yellow). The third one is the interaction between Au1 and two adjacent Au2 atoms (Fig. 5C, highlighted in yellow). The fourth one is the interaction between Au1, Au2 atoms, and S atom of L169C (Fig. 5D, highlighted in yellow). The electronic structure was unveiled by the localized orbital locator (LOL) profiles (Fig. 5E–H), which were defined by HL Schmider and AD Becke to characterize the electron localization[86]. In the regions surrounded by the isosurface of the LOL with a large value, electrons are more localized in them. Relatively, electrons can freely delocalize in such regions[87,88]. The highlighted LOL profile confirmed the four types of multi-center interaction between Au1/Au2/L169C (Fig. 5E–G). Fig. 5E, G evidenced the coordination nature between Au1 and R168H/L169C. The unassigned residual densities in the Fo-Fc difference maps in the 4-fold channel in the cluster structure might be related to the electron delocalization as indicated by the highlighted region among Au atoms (Fig. 5F–H, Supplementary Fig. S4).

To gain a better understanding of the deep nature of the Au-X (N, S) coordination bond, we extracted the crystal structure of the Cys48 site in Au(200 equiv.)-apo-R168H/L169C-rHLFr and performed quantum chemical calculations (see details in method section), followed by the calculation of orbital composition using Multiwfn based on natural bond orbital (NBO) analysis[89] and

natural atomic orbitals[90] The Supplementary Fig. S10 displays the molecular orbital (MO) that encompasses the atoms of interest. The examination of the orbital composition of Au and N reveals that the coordination bond between the two is primarily composed of 5d (Au) and 2p (N) orbitals. The major contributions to the Au–S bond come from the 5d (Au), 6 s (Au), and 3p (S) orbitals. This also supports the fact that being soft acid, gold ions have the tendency to bind to soft base, S(Cys or Met) according to HSAB principle. N (His) being borderline base can support the Au–S coordination as we observed in the crystal structures (Fig. 3).

## Conclusion

In this work, we have engineered and constructed a gold clustering site at the 4-fold symmetric axis channel of the apo-ferritin cage. X-ray crystal structures in combination with QC calculation revealed its unique and tunable structures with up to 12 Au atoms positions in the cavity. We evaluated the stepwise accumulation process of Au ions into the cage and the formation of a multinuclear Au cluster in our designed cavity. During the accumulation process, key residues such as His49, Glu45, Arg52, Glu130 were identified and their dynamic conformational changes were captured. The structure comparison of the clusters at different Au concentrations capitalized on the crucial role of the dynamics behavior R168H exhibited in configuration transformation. Therefore, it is expected to overcome the major challenge to design and precise tune their structures at the atomic level. In addition, the current work not only provides a method to prepare precise Au clusters in symmetric protein scaffold but also offers to design symmetric N, S donor organic ligands. Overall, the present demonstration is expected to be of great interest to design, control, and construct gold clustering sites in protein scaffolds.

## Methods

**Protein purification.** apo-R168H/L169C-rHLFr mutant DNA was prepared using the inverse PCR method. Recombinant L-chain apo-rHLFr from horse liver (rHLFr) was prepared by transfection of the expression vector pMK2 into the NovaBlue competent cells (Novagen). The cultivation and purification of the ferritin followed the same procedures as reported previously[37,38]. After Cell crushing and heat treatment, the supernatant was further purified by AKTA system with a Q sepharose High-Performance column (GE Healthcare, USA) and an S-300 column (GE Healthcare, USA).

**Preparation Au·apo-R168H/L169C-rHLFr.** 5 mL of 5 μM freshly purified protein was gently mixed with appropriate equivalents of chloro(dimethylsulfide) gold(I) solution from a 10 mM freshly prepared stock in acetonitrile. The reaction was proceeded with stirring at 20 C for 2 h. Then, the reaction mixture was put into dialysis against 50 mM pH 8.0 Tris-HCl buffer overnight. Finally, the unbound metal ions and organic ligand were further removed by size exclusion chromatography using a G-200/G-25 column. The purified metal composites were stored at 4 °C for further analysis.

**Crystallization.** The crystals of ferritin mutants and metal composites were obtained using a hanging-drop vapor diffusion method. The precipitant solution contains 0.5–1 M $(NH_4)_2SO_4$ and 10–20 mM $CdSO_4$. The drops were prepared by mixing 1.5 μL of concentrated ferritin solution (15-20 mg/mL) and 1.5 μL of the precipitant solution. The crystals formed within 24 h in the equilibration of the protein mixture and the precipitant solution at 20 °C.

**X-ray diffraction data collection.** The X-ray diffraction data of the ferritin crystals were collected at beamline of BL45XU at SPring-8 or X-ray crystallography facility at Tsinghua University (XtaLAB Synergy Custom FRX and a hybrid photon counting detector HyPix-6000, Rigaku, Japan). The crystals were soaked into the cryoprotectant containing 10% and 25% (w/w) glycerol sequent before data collection. Then, the crystals were feezed in liquid nitrogen. X-ray diffraction data were collected at 100 K. The data process was proceeded using HKL2000/Crysalis Pro programs with the space group of F432.

**Refinement of crystal structures.** The crystal structures of the apo-ferritin mutants and metal-ferritin composites were determined using the method of molecular replacement (MOLREP). The crystal structure of the apo-rHLFr wild type (PDB code: 1DAT) was used as an initial model. Data scaling was done using the AIMLESS program in CCP4. The structures were refined using the REFMAC5[91] program in the CCP4 package and modeled in COOT[92] using 2Fo–Fc and Fo–Fc electron-density maps. Water molecules were positioned to fit residual Fo-Fc density peaks with a cutoff of 3σ. The positions of the metal atoms were determined by the anomalous electron-density maps with a cutoff of 4σ. The anomalous map was calculated using the FFT program in CCP4. The Au atoms were distinguished from Cd by comparison with the control apo-R168H/L169C-rHLFr structure and the anomalous density maps at multiple wavelengths of 1.15 Å and 1.00 Å (Supplementary Fig. S3). The occupancies were determined by the manual adjustment considering the Fo–Fc difference maps and surrounding B-Factors. At the 4-fold channel, even after assigning 8–12 Au ions based on the anomalous map at 4σ, there remained additional residual densities in the Fo–Fc maps (Supplementary Fig. S4). Due to the lack of any anomalous density, it is hard to assign metals in those residual densities. In addition, they are very close to the Au ions and cover an extended area with an unusual shape which might be from water ligands. Therefore, we left unassigned and considered only the major Au binding positions based on anomalous density. Gln158, 169Cys (400 equiv. structure) and Lys172 (10 equiv. structure) were replaced with Ala, C-terminal residues Lys172-Asp174 (apo, 50 equiv., 100 equiv., 200 equiv., 400 equiv. structures) or His173-Asp174 (10 equiv., 200 equiv. structures) were not assigned due to the insufficient electron density for assignment. To ensure reproducibility, all experiments were repeated at least twice from protein expression to structure resolution. The structures were validated using the wwPDB validation server and PROCHECK in CCP4. The X-ray diffraction measurement parameters and refinement statics are listed in Table S1.

**ICP-MS/BCA.** Inductively coupled plasma mass spectrometry (7800 ICP-MS, Agilent, USA) and bicinchoninic acid assay (BCA) were applied to determine the Au and protein concentrations in apo-R168H/L169C-rHLFr Au composites. The Gold Standard for ICP (Sigma-Aldrich) was used as the standard for ICP-MS measurements.

**XPS.** X-ray photoelectron spectroscopy (XPS) spectra of apo-R168H/L169C-rHLFr Au composites acquired on a Thermo Scientific K-Alpha (Thermo Scientific, USA) using an Al Ka X-ray source (6 mA, 15 kV). The freshly prepared apo-R168H/L169C-rHLFr Au composites were concentrated to about 30 μM and dropped on clean Si plates to dry, then used for XPS analysis. The binding energy was corrected using C1s spectra (284.8 eV) as a standard before further analysis.

**UV-vis spectroscopy.** The UV-visible absorption spectral measurements were performed using UV-2700 UV-Visible Spectrophotometer (Shimazu, Japan).

**EEM spectroscopy.** The Excitation Emission Matrix (EEM) spectroscopy measurements were performed on a Horiba Aqualog fluorescence spectrometer (Horiba Aqualog 800-C, Horiba, USA).

**Computational details.** The quantum mechanics calculation were performed on the Gaussian 16 program with the PBE0 hybrid density functional method. D3-BJ version of Grimme's empirical method was applied to correct the dispersion effects in all calculations. The geometry optimizations were carried out using the 6-311 G(d) basis set for C, N, S, H atoms and Stuttgart/Dresden (SDD) pseudopotential for Au atoms. During the geometry optimization, the positions of Au ions and the C-alpha atoms of coordinating residues were fixed since their electron density were clearly observed. Frequency calculations were conducted to obtain the thermal corrections to Gibbs free energy at the same level. Single point calculations on the optimized structures were performed with the larger basis set def2tzvp to get more accurate electronic energies (with SMD solvation and D3-BJ dispersion effect correction). The values presented in this study thus the large basis set energies corrected for thermal corrections and solvation effects. The wavefunction analysis was conducted on Multiwfn[90].

**Visualization.** The figures of crystal structures were made in PyMol[93] or VMD[94]. The localized orbital locator (LOL) filled-color map was made in Multiwfn.

## Data availability

The atomic coordinates and structure factors of apo-R168H/L169C-rHLFr, Au(10 equiv.)-apo-R168H/L169C-rHLFr, Au(50 equiv.)-apo-R168H/L169C-rHLFr, Au(100 equiv.)-apo-R168H/L169C-rHLFr, Au(200 equiv.)-apo-R168H/L169C-rHLFr, Au(400 equiv.)-apo-R168H/L169C-rHLFr and Au(200 equiv.)-apo-R168C/L169C-rHLFr have been deposited in the Protein Data Bank, https://www.wwpdb.org/ (PDB ID codes 7VIO, 7VIP, 7VIQ, 7VIR, 7VIS, 7VIT, and 7VIU). The atomic coordinates of the optimized computational models in QC calculations were provided in Supplementary Data 1.

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

## Acknowledgements

This study was financially supported by the National Key Research and Development Program of China under Grant No. 2018YFA0902200 and the Chinese National Natural Science Foundation under Grant No. 21878175. We thank the Tsinghua University Branch of China National Center for Protein Sciences (Beijing) and Tsinghua University Technology Center for Protein Research for the X-ray Crystallography Facility facility support. This work was supported by funds from the Tsinghua University Branch of China National Center for Protein Sciences (Beijing).

## Author contributions

C. L.: conceptualization, investigation, methodology, X-ray data collection, structure refinement, QC calculations, data analysis, visualization, writing—original draft and editing and revision. B. M.: methodology, structure refinement, crystal structure supervision, writing—editing and revision. X. P.: investigation, X-ray data collection, structure refinement, data analysis, visualization. N. I.: plasmid preparation. X-ray data collection. S. A.: structure refinement, crystal structure supervision. X. S.: QC data supervision. T. U.: project administration, supervision, crystal structure supervision, writing—review & editing. D. L.: project administration, funding acquisition, supervision, writing—review and editing.

## Competing interests

The authors declare no competing interests.
