## [Peer Review File · Communications Chemistry]

Reviewers' comments:

Reviewer #1 (Remarks to the Author):

In this manuscript, Lu et al. report the X-ray crystal structures of Au(I) associated Apo-ferritin cages at varied dosages of Au(I) precursors. In order to facilitate the adsorption of Au(I), the authors employed a mutant Apo-ferritin containing 4 L169Cs and R168Hs at its 4-fold channel (i.e., Apo-R168H/L169C-rHLFr) as the model protein, where the Cys and His residues are expected to offer good bonding affinity to Au(I). The structures of Apo-R168H/L169C-rHLFr after its reaction with varied dosages of Au(I) were revealed by X-ray crystallography, manifesting 4 types of bonding sites for Au(I). Among these bonding sites, the 4-fold bonding sites are new to the community. The most interesting findings of this work is that the amount and structure of Au(I) on the 4-fold bonding sites are dosage-dependent. Also, based on the crystallography data, the authors map out the affinity sequence of the bonding sites to Au(I). The interactions of metal ions with proteins are attracting recent interest, as they are not only crucial in biomedical industry, but also important for the biological or biomimetic synthesis of metal nanomaterials. I therefore believe the fundamental insights on the bonding sites and bonding affinity of Au(I) in proteins should be of interest to heterogeneous readers, particularly in the fields of noble metal chemistry, cluster chemistry, supramolecular chemistry, and many relevant applications. I hence would like to recommend acceptance of this manuscript for publication after necessary minor revisions (see below for my detailed comments).

- 1) How stable are the Au(I)-Apo-R168H/L169C-rHLFr composites in solution? Will the content of Au(I) change with time going by?
- 2) I noted that the authors spend less lines to describe the morphology of the Au₁₂ cluster on the 4-fold site. Are they icosahedral in shape? Icosahedral M₁₂ or centered-icosahedral M₁₃ (M denotes Au or Ag) are common kernel structure observed in thiolate protected Au or Ag nanoclusters.
- 3) The structure change of Au₁₂ cluster induced by the dosage of Au(I) on the 4-fold site is interesting. What is the driving force for such structure change? The underlying chemistry governing such structure change rather than the phenomenon itself should be more attractive to the readers.
- 4) What is the possible application of these Au(I)-Apo-R168H/L169C-rHLFr composites? The authors may like to present their UV-vis absorption and photoluminescent spectra, which may provide hints for their potential applications.
- 5) Can the Au(I)-Apo-R168H/L169C-rHLFr composites be converted to Au(0) nanoclusters by reduction?
- 6) There are many typos or editorial errors which are annoying. The authors need to prepare their manuscript with greater care. Some examples are (this is not a complete list):
 - a. Line 62: "reocnstructed" should be "reconstructed".
 - b. Line 107: what does "relatively single" mean?
 - c. Line 126: there is a repeating word "showing".
 - d. Many unidentified symbols in Lines 140, 235, 240, etc.

Reviewer #2 (Remarks to the Author):

In this paper, the authors constructed a novel gold clustering site at the 4-fold symmetric axis channel of the apo-ferritin cage. The stepwise accumulation process of Au ions into the cage and the

formation of gold clusters in the cavity were evaluated. The role of key residues in the metal accumulation process was disclosed. As a result, the unique Au clustering site with up to 12 Au atoms in the cavity was obtained, and such a structure of the Au cluster can be designed. The binding affinity order of the four Au binding sites on apo-ferritin was further unveiled. The construction of this apo-ferritin cage and Au clusters in the cage are interesting, and the manuscript is well written with solid data to support their conclusions. I believe this work will be of interest to a broad scientific audience, especially for cluster material scientists. Thus I would like to suggest the acceptance of this paper after the authors have addressed the following minor issues.

(1) In the X-ray crystal structures of these gold nanoclusters, Au atoms are anchored into the designed cage via different Au-X interactions, including Au-N, Au-Cl, Au-S (and Au-C?). What is the underlying chemistry among these interactions (such as the interaction robustness, the reactivity, etc)?

(2) Is that possible to evaluate the entry route of Au atoms to this cage? For example, as depicted in Figure 2A, are Au atoms first anchored onto several peripheral positions, and then transferred to the inner 4-fold symmetric cage? Or just occupy the 4-fold symmetric cage directly?

(3) In previously reported protein-stabilized gold or silver nanoclusters, the metal clusters are always emissive. Thus, are the Au clusters (especially, the Au₁₂ cluster) in the designed cage emissive? If so, what is the relationship between the emission and the number of Au atoms in clusters?

Reviewer #3 (Remarks to the Author):

Lu et al report an engineering and construction of a new-to-nature gold clustering site at the 4-fold symmetric axis channel of the apo-ferritin cage. The stepwise accumulation of Au(I) into the cage and the formation of Au clustering (up to 12 atoms) in the designed cavity are evaluated by a series of X-ray crystal structures. QC calculations are further carried out to obtain insights.

This work is quite interesting, and the results should benefit other researchers in the community. I suggest its acceptance for publication in CommsChem.

Improvements:

The figures are very far away from the paragraphs that discuss them! Pls move them to where they are first mentioned in the main text.

Can the authors provide the UV-vis spectra of 10/50/100/200/400 equiv. samples (before and after crystallization)?

Language errors: "have showed Cd binding": change showed to shown.

A point-by-point response to the reviewers' comment

Reviewer #1

The interactions of metal ions with proteins are attracting recent interest, as they are not only crucial in the biomedical industry but also important for the biological or biomimetic synthesis of metal nanomaterials. I, therefore, believe the fundamental insights on the bonding sites and bonding affinity of Au(I) in proteins should be of interest to heterogeneous readers, particularly in the fields of noble metal chemistry, cluster chemistry, supramolecular chemistry, and many relevant applications. I hence would like to recommend acceptance of this manuscript for publication after necessary minor revisions (see below for my detailed comments).

Reply: Thanks for the reviewer's positive comment on this work.

(1) How stable are the Au(I)-Apo-R168H/L169C-rHLFr composites in solution? Will the content of Au(I) change with time going by?

Reply: We appreciate the reviewer's question which we think, we should have checked this earlier. To examine the stability, we measured the absorption spectra of Ferritin-Au composites at various time intervals of 1 day, 3 days, and 5 days, because the reduction of Au ions can be detected in the absorption spectrum. We found no noticeable changes in the spectra. This concludes that our composites (1-5 μM) are stable for at least the above mentioned time periods. However, after a long time (over 2 weeks) the crystals and high concentration solution ($\sim 40 \mu\text{M}$) utilized for crystallization gave naked eye-visible color changes. The results are now included in the revised manuscript (Page 8, line 141-148, Figure S8).

To inspect any changes in the Au content with time, we quantitatively measured the number of Au ions per cage after 5 days of preparation using the ICP/BCA method. No significant changes were observed for Au(200equiv)-apo-R168H/L169C-rHLFr as the number of bounded Au atoms are 139 and 137 for 4 $^{\circ}\text{C}$ and 25 $^{\circ}\text{C}$ samples, respectively. We have now included the results in the revised manuscript (Table S2).

(2) I noted that the authors spend less lines to describe the morphology of the Au₁₂ cluster on the 4-fold site. Are they icosahedral in shape? Icosahedral M₁₂ or centered-icosahedral M₁₃ (M denotes Au or Ag) are common kernel structures observed in thiolate protected Au or Ag nanoclusters.

34 **Reply:** Thank you very much for your question. As the reviewer pointed out, we have now
35 discussed this in the revised manuscript (Page 16, line 314-319). We agree with the reviewer
36 that Icosahedral or centered-icosahedral are common kernel structures observed in thiolate-
37 protected Au₁₂ and Au₁₃ nanoclusters. In our case, the asymmetric protein environment leads
38 to giving the unique facial square shaped structure which is uncommon in literature. As
39 demonstrated in Figure 4, the Au₁₂ cluster on the 4-fold site exhibits a two-layer layered
40 morphology. It highlights the unique channel topologies that aided in the creation of this one-
41 of-a-kind Au clustering site.

42

43 *(3) The structure change of the Au₁₂ cluster induced by the dosage of Au(I) on the 4-fold site*
44 *is interesting. What is the driving force for such structure change? The underlying chemistry*
45 *governing such structural change rather than the phenomenon itself should be more*
46 *attractive to the readers.*

47 **Reply:** We thank the reviewer for pointing out this interesting fact. Actually, we discussed a
48 little about this in the previous manuscript (Page 17, line 319-327). As pointed out, we
49 discussed more about the driving force of the structure change etc. in the revised manuscript
50 (Page 17, line 327-333). At higher gold precursor concentration, the equilibrium of metal-
51 ligand (protein) interaction was changed. Since amino acid side chains are flexible and can
52 change their conformations according to the metal coordination structure, a more
53 energetically favourable Au cluster structure was thus formed at 400 equiv. Figure 3C shows
54 the conformational changes of L169H with increasing Au precursor concentrations which
55 were reflected in the overall cluster structure in the 4-fold channel as well as structural
56 transition.

57

58 *(4) What is the possible application of these Au(I)-Apo-R168H/L169C-rHLFr composites?*
59 *The authors may like to present their UV-vis absorption and photoluminescent spectra, which*
60 *may provide hints for their potential applications.*

61 **Reply:** Thank you very much for your question. Gold nanoclusters have wide applications
62 starting from imaging to catalysis and other materials applications. The most difficult part is
63 how to design, control and prepare the desired cluster? which is actually the goal of the
64 current manuscript. However, our Ferritin-Au(I) composites might be useful for the following:
65 (i) Au(I) catalysis taking the advantage of the asymmetric protein environment. We are
66 currently working on that. (ii) If we can control the number of Au ions to have luminescent
67 properties, then, it can be used for imaging or other biomedical applications. (iii) Exploring

68 the fundamental mechanism of gold nanocluster formation by addition of reducing agent
69 followed by structure determination.

70 We have now included the UV-visible and photoluminescent spectra in the revised
71 manuscript (Page 20, line 392-397, Figure S8-S9).

72

73 (5) *Can the Au(I)-Apo-R168H/L169C-rHLFr composites be converted to Au(0) nanoclusters*
74 *by reduction?*

75 **Reply:** Thank you very much for raising this question. It is possible to convert the Au(I)
76 composites to Au(0) by addition of reducing agent and the changes can be measured by X-ray
77 crystal structure analysis. We did similar work using wild-type ferritin before (*Nat. Commun.*
78 *2017, 8, 1–9*). However, such an experiment is beyond the goal of this manuscript and might
79 be published separately.

80

81 (6) *There are many typos or editorial errors which are annoying. The authors need to*
82 *prepare their manuscript with greater care. Some examples are (this is not a complete list):*

83 *a. Line 62: “reocnstructed” should be “reconstructed”.*

84 *b. Line 107: what does “relatively single” mean?*

85 *c. Line 126: there is a repeating word “showing”.*

86 *d. Many unidentified symbols in Lines 140, 235, 240, etc.*

87 **Reply:** Thank you so much for your careful check. We corrected all.

88

89

90 **Reviewer #2**

91 *The construction of this apo-ferritin cage and Au clusters in the cage are interesting, and the*
92 *manuscript is well written with solid data to support their conclusions. I believe this work*
93 *will be of interest to a broad scientific audience, especially for cluster material scientists.*
94 *Thus I would like to suggest the acceptance of this paper after the authors have addressed the*
95 *following minor issues.*

96 **Reply:** Thanks to the reviewer for his/her positive comments.

97

98 (1) *In the X-ray crystal structures of these gold nanoclusters, Au atoms are anchored into the*
99 *designed cage via different Au-X interactions, including Au-N, Au-Cl, Au-S (and Au-C?).*

100 *What is the underlying chemistry among these interactions (such as the interaction*
101 *robustness, the reactivity, etc)?*

102 **Reply:** Thank you very much for the comment. We have included an explanation for Au-
103 X(protein) interaction in the revised manuscript (Page 23, line 447-457). For a better
104 understanding of the Au-X bond, we also performed additional QM calculations. The results
105 are discussed in the revised manuscript with an additional figure (Page 23, line 447-455,
106 Figure S10).

107

108 *(2) Is that possible to evaluate the entry route of Au atoms to this cage? For example, as*
109 *depicted in Figure 2A, are Au atoms first anchored onto several peripheral positions, and*
110 *then transferred to the inner 4-fold symmetric cage? Or just occupy the 4-fold symmetric*
111 *cage directly?*

112 **Reply:** We are quite appreciative of your insightful comment. The route and entry
113 mechanism of Au ions into the cage is already given in Figure 3, Figure S2 and Figure S5.
114 The Au ions enter into the cage through the 3-fold channel as we first observed Au binding
115 here at low Au concentration. As pointed out, we are now emphasizing more on this in the
116 revised manuscript (Page 13, line 261-269).

117

118 *(3) In previously reported protein-stabilized gold or silver nanoclusters, the metal clusters*
119 *are always emissive. Thus, are the Au clusters (especially, the Au₁₂ cluster) in the designed*
120 *cage emissive? If so, what is the relationship between the emission and the number of Au*
121 *atoms in clusters?*

122 **Reply:** We agree with the reviewer that there are several reports on emissive protein
123 stabilized gold nanoclusters. To check whether our composites are emissive or not, we
124 measured Excitation Emission Matrix (EEM) spectroscopy using our Ferritin-Au(I)
125 composites. None of them showed any emissions. We searched the literature and found that
126 the majority of the emissive gold clusters contained Au(0) oxidation state. Therefore, it is not
127 unusual that our Au(I) clusters are non-emissive. In the revised manuscript, we included both
128 the results and explanation (Page 20, line 392-397, Figure S9).

129

130 **Reviewer #3**

131 *This work is quite interesting, and the results should benefit other researchers in the*
132 *community. I suggest its acceptance for publication in CommsChem.*

133 **Reply:** Thanks to the reviewer for his/her positive comments.

134

135 *Improvements:*

136 (1) *The figures are very far away from the paragraphs that discuss them! Pls move them to*
137 *where they are first mentioned in the main text.*

138 **Reply:** We agree with the reviewer that the figure position and discussion should be nearby.
139 We think it will be adjusted during the conversion of this manuscript into “Commun Chem”
140 format.

141

142 (2) *Can the authors provide the UV-vis spectra of 10/50/100/200/400 equiv. samples (before*
143 *and after crystallization)?*

144 **Reply:** Thank you very much for your constructive suggestions. We are now including the
145 absorption spectra of all the composites in the solution in the revised manuscript (Figure S8).
146 Measuring the absorption after crystallization (in solid state) is difficult and it needs a special
147 instrument. In addition, UV-visible spectra of the solid crystal might be informative, but
148 beyond the goal of this manuscript. Thus, we did not measure it. As an alternate, we
149 measured the absorption spectra at various time intervals in solution to check the stability of
150 the Ferritin-Au(I) composites. The results are included in the revised manuscript (Page 8, line
151 141-148, Figure S8).

152

153 (3) *Language errors: "have showed Cd binding": change showed to shown.*

154 **Reply:** Thank you so much for your careful check. We corrected the mistake.

155

REVIEWERS' COMMENTS:

Reviewer #1 (Remarks to the Author):

The authors have addressed my remaining concerns properly. Unfortunately, the Au(I) clusters produced in this work are not emissive. However, this is surely not the focus of current work. Therefore, I support publication of this manuscript as it is. Congratulations to the authors on their achievements.

Reviewer #2 (Remarks to the Author):

The authors have revised the manuscript according to the comments of the reviewers. The quality of the manuscript has been greatly improved. I think it is now acceptable.